# Combination of Photodynamic Therapy and Oral Antifungals for the Treatment of Onychomycosis

**DOI:** 10.3390/ph15060722

**Published:** 2022-06-07

**Authors:** Alba Navarro-Bielsa, Tamara Gracia-Cazaña, Pilar Robres, Concepción Lopez, María Dolores Calvo-Priego, Carmen Aspiroz, Yolanda Gilaberte

**Affiliations:** 1Department of Dermatology, Miguel Servet University Hospital, IIS Aragón, Universidad de Zaragoza, 50009 Zaragoza, Spain; tamgracaz@gmail.com (T.G.-C.); concepta963@gmail.com (C.L.); lolicalvo@hotmail.es (M.D.C.-P.); ygilaberte@gmail.com (Y.G.); 2Microbiology Unit, Barbastro Hospital, 22300 Huesca, Spain; pilarrobres7@gmail.com; 3Microbiology Unit, Royo Villanova Hospital, 50015 Zaragoza, Spain; carmenaspiroz@gmail.com

**Keywords:** onychomycosis, photodynamic therapy, oral antifungals

## Abstract

Onychomycosis accounts for 50% of nail disorders, making it one of the most prevalent fungal diseases and a therapeutic challenge. Photodynamic therapy (PDT) could constitute a therapeutic alternative, owing to its good adherence, the low probability of resistance, the lack of interaction with antimicrobials, and its favorable adverse effect profile. This retrospective observational study included all patients with a microbiological diagnosis of onychomycosis treated with PDT at Miguel Servet University Hospital, Zaragoza (Spain), between January 2013 and June 2021. The protocol consisted of pre-treatment with 40% urea for 7 days, followed by 16% methyl-aminolevulinate (MAL) for 3 h and subsequent irradiation with a red-light LED lamp (37 J/cm^2^), every 1 or 2 weeks. Combined treatment with oral and/or topical antifungals was recorded. Of the 20 patients included (mean age, 59 ± 17 years), 55% were men. The most frequently detected microorganism was *Trichophyton rubrum* (55%). The most commonly affected location was the feet (90%): 50% of these cases were associated with tinea pedis. The median (standard deviation) number of PDT sessions was 6 (2.8). PDT was combined with systemic terbinafine (250 mg/day) in 10 cases (in 8 cases, this was administered for only 1 month), and with topical terbinafine in 3 cases. A complete clinical response was achieved in 80% (16) of cases and microbiological cure in 60% (12). PDT is a therapeutic alternative for onychomycosis, and can be administered either in monotherapy or combined with antifungals, allowing for a reduction in the duration and possible adverse effects of antifungal treatment and achieving higher cure rates than those obtained with either treatment alone.

## 1. Introduction

Onychomycosis accounts for 50% of nail disorders, making it one of the most prevalent fungal diseases [1]. It is one of the most therapeutically challenging superficial mycoses, with treatment failure rates ranging from 10 to 53%. While the antifungal terbinafine provides the best cure rates, it requires prolonged administration, can cause adverse effects, and suffers from pharmacological deficiencies, including limited penetration capacity and the development of resistance [2,3].

Photodynamic therapy (PDT) may constitute a useful therapeutic alternative for superficial skin infections, including onychomycosis. The advantages of this modality include a short treatment duration, which favors adherence; a multitarget mechanism of action, which implies a low probability of developing resistance; a lack of drug interaction, which facilitates possible combination with antifungals; and its being easily reproducible, and having mild adverse effects with an acceptable risk–benefit profile [4,5].

The aim of this study was to examine the effectiveness of PDT combined with antifungal treatment in patients with onychomycosis, and to evaluate the adjuvant effect of PDT in this context, which may allow for a reduced duration of systemic antifungal treatment.

## 2. Results

A total of 20 patients were included in our analysis. The characteristics of the study population are presented in Table 1. Males accounted for 55% of the population, and the mean (±SD) age was 59.4 ± 17.12 years (range, 13–86 years). The most frequently detected microorganism was *Trichophyton rubrum* (11 patients, 55%), followed by *Aspergillus terreus* (3 patients, 15 %), and the most frequently affected location was the feet (90%). Half of all cases of foot involvement were associated with tinea pedis. The number of nails affected was 1 in 6 patients (30%), 2 in 8 patients (40%), and 3 or more in 6 patients (30%). Seven patients (35%) had not received previous treatment, 9 (45%) had undergone topical antifungal treatment, and 4 (20%) systemic antifungal treatment.

The median (±SD) number of PDT sessions was 6 (2.8), with a range of 3–15 sessions. In all cases, MAL was used as a photosensitizer (although in one patient this was switched to methylene blue [MB] after three sessions), and 40% urea was applied 7 days before PDT. None of them had painful sensation during the sessions. Ten patients (50%) received concomitant systemic terbinafine treatment (8 for only 4 weeks and 2 for 12 weeks) and three patients (15%) received concomitant topical terbinafine treatment.

After 6 months of follow-up, 80% showed a clinical response and 60% achieved microbiological cure (Figure 1). Sixteen patients (80%) achieved clinical resolution, 12 patients (60%) showed a complete clinical and microbiological response, and 4 (20%) improved clinically despite persistent positive culture results.

In the group that received combined treatment (Table 2), 7 of the 10 patients treated with PDT + systemic terbinafine showed a clinical response, of whom 5 achieved microbiological cure, and 3 of the 3 patients treated with PDT + topical terbinafine showed a clinical response, of whom 2 achieved microbiological cure. Dermatophyte infections were recorded in 11 patients and mold infections in 2 (Table 3), with response rates of 73% and 50%, respectively. Of 9 patients (45%) with tinea pedis, 3 (15%) had a positive culture after treatment, despite receiving adjuvant antifungal treatment.

## 3. Discussion

The efficacy of using PDT to treat onychomycosis, both in monotherapy and combined with antifungals, is well demonstrated, expanding the therapeutic arsenal for onychomycosis and helping to overcome some of the limitations of oral and systemic antifungal treatment [5,6]. The present findings show that the combination of PDT + terbinafine improves upon the results obtained with PDT alone, allowing for a reduction in the duration of systemic treatment and providing better outcomes in recalcitrant cases.

Despite advances enabled by the development of new antifungal compounds, onychomycosis treatment failure and recurrence are frequent. To date, the highest mycological response rates were achieved with terbinafine (250 mg/day) for 4 months and with itraconazole administered as pulse therapy or at 200 mg/day for 3–4 months, with response rates of 76%, 63%, and 59%, respectively [7]. Cure rates reported for topical antifungals include 38–54% for amorolfine (5%) administered twice per week for 6 months, 70% for tioconazole (28%) administered twice daily for 6–12 months, and 28–36% for ciclopirox olamine (8%) administered daily for 48 weeks [8]. However, topical antifungals are only effective for distal and lateral or superficial onychomycosis without dystrophy. By contrast, antifungals show poor efficacy for the treatment of non-dermatophyte molds [9].

Studies have demonstrated the effectiveness of PDT in onychomycosis patients using different photosensitizers, such as aminolevulinate (ALA) [10], MAL [6], MB [11] and rose Bengal [12], curcumin [13], and compared its efficacy in monotherapy with available drugs [14]. PDT appears to be more effective than amorolfine for the treatment of non-dermatophyte onychomycosis [15]. Four clinical trials have compared PDT with oral antifungals for the treatment of onychomycosis. Using 20% ALA and red light, Sotiriou et al. [16] treated 30 patients with *T. rubrum* onychomycosis, of which 73.3% showed involvement of the first toenail, and reported a response rate of 43% after 12 months of follow-up. Tardivo et al. [17] reported similar results in 62 cases of onychomycosis caused by *Trichophyton rubrum*, *T. mentagrophytes*, or *Candida albicans.* In that study, MB or toluidine blue were used as photosensitizers, and after 5 min, the nail was superficially irradiated at a distance of 5 cm for 3 min with a light in the red region of the spectrum, resulting in a final irradiance of 18 J/cm^2^, with complete clearance achieved in 45% of patients. Figueiredo Souza et al. [18,19] conducted two studies using 2% MB as a photosensitizer. The first trial, which included 22 patients divided into two groups according to severity, revealed a significantly better clinical response in patients with mild-to-moderate (100%) onychomycosis versus those with severe onychomycosis (63.6%) [18]. In the second, larger trial, 80 patients with onychomycosis caused by dermatophytes, yeasts, and non-dermatophyte molds were randomized to receive either MB-PDT every 2 weeks until clinical cure or oral fluconazole (300 mg per week for 24 weeks). Mechanical abrasion of the nail plate was performed prior to PDT to facilitate penetration of the photosensitizer. On completion of treatment, the cure rate was 90% in the MB-PDT-treated group versus 45% in the oral fluconazole group (*p* < 0.002) [19]. Our team previously conducted a randomized clinical trial to investigate the efficacy and safety of MAL-PDT in onychomycosis patients. A total of 40 patients underwent three sessions of urea (40%) treatment for 7 days, followed by (i) conventional 16% MAL-PDT or (ii) urea (40%), followed by red-light irradiation [6]. MAL-PDT was significantly superior to red-light PDT only for the treatment of non-dystrophic onychomycosis. Only one study compared the combination of PDT and antifungals; specifically, oral terbinafine combined with nine sessions of either MB-PDT or MAL-PDT every 2 weeks for 12 weeks, with the application of 40% urea between sessions, in 20 patients with severe dermatophyte onychomycosis of the big toe. Microbiological cure was achieved in 100% of patients in the MB-PDT group versus 90% in the MAL-PDT group, and complete cure was achieved in 70% of patients in both groups, indicating that PDT is an effective method to accelerate terbinafine-mediated healing [11].

The present findings indicate a good rate of clinical resolution (80%), similar to the highest rate reported in other studies [17]. We were prompted to investigate the effects of antifungals combined with PDT in onychomycosis patients based on our previous finding [6] that the proportion of onychomycosis patients reinfected after PDT was higher in those with dermatophyte versus mold onychomycosis. The most plausible explanation for this observation is that dermatophytes are largely found in the nails and on the feet; therefore, the addition of an antifungal provides a better outcome, since PDT only targets the nail. By contrast, in the case of non-dermatophyte onychomycosis, the causative microorganism typically resides exclusively in the nail; therefore, adjuvant treatment is less necessary [20]. Comparing the present findings with our previous clinical trial [6], 70% of patients with dermatophyte onychomycosis treated with PDT + terbinafine achieved a clinical response, compared with only 53.85% of those that received PDT only, which supports our hypothesis. Among patients with mold onychomycosis, only 50% of those treated with PDT + terbinafine achieved a clinical response, as compared with 67% of those that received PDT only in our previous clinical trial. This suggests a poor response of molds to antifungal treatment, and points to PDT alone as the preferred treatment for this type of microorganism [21].

Based on the present findings, and those of previous studies [6], we propose a treatment protocol combining PDT and oral terbinafine for the treatment of dermatophyte onychomycosis (Figure 2), consisting of three MAL-PDT sessions separated by 1 or 2 weeks and concomitant terbinafine treatment (250 mg/day) for 1 month, followed by another cycle of three sessions of MAL-PDT if necessary (i.e., if culture continues to be positive). The protocol could be repeated as many times as needed. Our group is currently working on the validation of this protocol in clinical practice.

## 4. Methods and Materials

We conducted a retrospective observational study of all patients with onychomycosis treated with PDT at Miguel Servet University Hospital, Zaragoza (Spain), between January 2013 and June 2021. All patients had a microbiological diagnosis and some of them had been previously treated with oral and/or topical antifungals without success.

Treatment consisted of 3-h MAL-PDT sessions (Metvix, Galderma, France) under occlusion and subsequent irradiation for 7–9 min (37 J/cm^2^, λ = 630 nm; Aktilite CL 128, PhotoCure ASA, Oslo, Norway), with a distance between the lamp and the treatment area of 5–6 cm, separated by 1 or 2 weeks. Nail plates were softened with 40% urea ointment 7 days before PDT sessions to enhance the penetration of the photosensitizer.

Data on concomitant oral antifungal treatment, and the duration thereof, were also recorded.

Disease involvement was determined at baseline and at the end of the study by taking clinical photographs and microbiological samples for direct examination and performing fungal cultures of samples taken from the nails and feet.

Clinical and microbiological effectiveness were assessed at week 12 (main outcome) and reevaluated at week 24.

### Statistical Analysis

Quantitative variables are expressed as mean and standard deviation (SD) and dichotomous variables as proportions. Associations between qualitative variables were assessed using Pearson’s Chi-squared test or Fisher’s exact test. The Wilcoxon test or Student’s *t*-test for paired data were used to evaluate associations between quantitative variables. Pearson’s correlation coefficient was calculated to evaluate the linear correlation between two variables. Statistical significance was set at *p* ≤ 0.05. Analyses were performed using SPSS Statistics (version 19.0: IBM, Armonk, NY, USA).

All collected data were recorded anonymously, strictly observing relevant data protection laws and regulations (Organic Law 3/2018 of 5 December). The study was approved by the Clinical Ethical Committee of Aragón (C.P.-C.I. EC12/0073).

## 5. Conclusions

In conclusion, PDT is a valid alternative for treatment of onychomycosis, either in monotherapy or combined with systemic or topical terbinafine, allowing for a reduction in the required dose of systemic antifungals, decreasing the likelihood of adverse effects and interactions, and achieving even higher cure rates, especially in patients with dermatophyte onychomycosis. Further clinical trials will be necessary to establish the optimum protocol for dermatophyte and mold onychomycosis.

## Figures and Tables

**Figure 1 pharmaceuticals-15-00722-f001:**
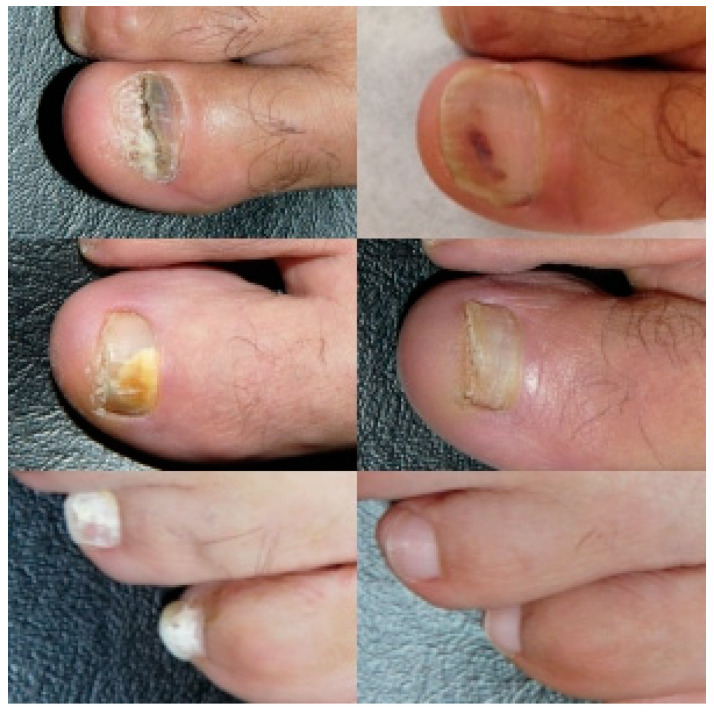
Clinical images of 3 patients before and after combined treatment with terbinafine and photodynamic therapy.

**Figure 2 pharmaceuticals-15-00722-f002:**
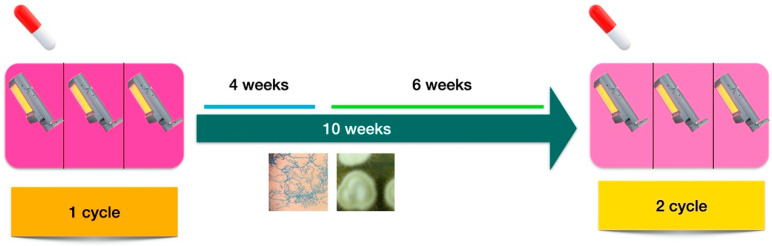
Protocol for photodynamic therapy combined with oral terbinafine for the treatment of dermatophyte onychomycoses. Three sessions separated by 1 or 2 weeks of MAL-PDT combined with terbinafine (250 mg/day) for 1 month, followed by another cycle of 3 sessions of MAL-PDT if necessary (i.e., in cases of persistent positive culture results).

**Table 1 pharmaceuticals-15-00722-t001:** Characteristics of the study population.

VARIABLE		N (%) *
AGE, MEAN ± SD [RANGE]		59.4 ± 17.12 [13–86]
SEX	Male	11 (55)
	Female	9 (45)
MICROORGANISM *T. RUBRUM* *A. TERREUS* *T. MENTAGROPHYTES* *A. SYDOWII* *A. FUMIGATUS* *F. OXYSPORUM*		11 (55) 3 (15) 2 (10) 2 (10) 1 (5) 1 (5)
LOCATION	Feet + Tinea pedis	18 (90) 9 (45)
	Hands	2 (10)
NUMBER OF NAILS INVOLVED	1	6 (30)
	2	8 (40)
	3 or more	6 (30)
NO. OF PDT SESSIONS, MEDIAN (±SD) [RANGE]		6 ± 2.8 [3–15]
PREVIOUS TREATMENT	No	7 (35)
	Topical antifungal	9 (45)
	Systemic antifungal	4 (20)
PHOTOSENSITIZER	Methyl aminolevulinate	20 (100)
PRIOR UREA APPLICATION		20 (100)
CONCOMITANT TREATMENT	No	7 (35)
	Topical terbinafine	3 (15)
	Systemic terbinafine - *1 month, {resolution}* - *3 months, {resolution}*	10 (50) 8 (40), {5 (25)} 2 (10), {0}
RESULTS	Microbiological and clinical resolution	12 (60)
	Clinical resolution	4 (20)
	Persistence	4 (20)

* Unless otherwise indicated.

**Table 2 pharmaceuticals-15-00722-t002:** Cure rates.

	PDT N (%)	PDT + Topical Terbinafine N (%)	PDT + Systemic Terbinafine, 1 Month N (%)	PDT + Systemic Terbinafine, 3 Months N (%)	Total N (%)
Clinical resolution	6 (30)	3 (15)	7 (35)		16 (80)
Clinical and microbiological resolution	5 (25)	2 (10)	5 (25)		12 (60)
Persistence	1 (5)		1 (5)	2 (10)	4 (20)

**Table 3 pharmaceuticals-15-00722-t003:** Microbiological cure after PDT alone or combined with terbinafine, according to causative micro-organism.

	Before PDT N (%)	After PDT N (%)	Before PDT + Topical Terbinafine N (%)	After PDT + Topical Terbinafine N (%)	Before PDT + Oral Terbinafine N (%)	After PDT + Oral Terbinafine N (%)
*T. rubrum*	11 (55)		2 (10)	2 (10)	8 (40)	5 (25)
*A. terreus*	3 (15)	1 (5)			2 (10)	1 (5)
*T. mentagrophytes*	2 (10)	1 (5)	1 (5)	1 (5)		
*A. sydowii*	2 (10)	2 (10)				
*A. fumigatus*	1 (5)	1 (5)				
*F. oxysporum*	1 (5)	1 (5)				

## Data Availability

Data is contained within the article.

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
