# Peer review of "Combination of Photodynamic Therapy and Oral Antifungals for the Treatment of Onychomycosis"

_pharmaceuticals, 2022, doi:10.3390/ph15060722_

Round 1

Reviewer 1 Report

After long years of follow-up, this study reports the importance of effective and combined treatment of terbinafine and photodynamic therapy with topical MAL.
I have few suggestions and some questions:
In the summary, correct the fluence unit (37 J/cm2).
Inform if it was collected and pain scale during lighting and/or, there is no painful sensation with nail treatment using Metvix.
Explain why one of the patients stoped the MAL treatment and was moved for a MB protocol.
Explain in more detail the characteristics of the equipment used (Aktilite).

  • What intensity, time and distance between LEDs and nails.
    As the size of the LED board is more than 15 cm long, how could they keep only the nails lit?
  • Have you had any cases where you had to interrupt the lighting due to discomfort?
  • How did you determine that 3 hours was sufficient for penetration of the cream and production of PpIX by the hyphae?
  • They presented published cases using other photosensitizers but did not mention the use of curcumin and synthetic porphyrin in the treatment of onychomycosis, which also present excellent clinical results and are highlighted in the clinical protocols of PDT.

Fig 1 does not report how many PDT sessions were needed to achieve those clinical results.
Would you recommend topical or systemic terbinafine?

Author Response

Thank you very much for your helpful comments.

All mistakes have been corrected and highlighted in red in the main text.

  • A pain scale was collected in all patients and none of them felt pain during the sessions, we did not stop any session due to this cause. “None of them had painful sensation during the sessions.” Added in line 89.
  • One of the patients stopped MAL treatment and was moved for a MB protocol because the patient preferred to discontinue treatment.
  • The model of equipment has been completed. “Aktilite CL 128” has added in line 53.
  • The nails were irradiated with a 635-nm LED lamp at a fluence of 37J/ cm2 for 7-9 minutes (37 j/cm2) and with a distance between the lamp and the treatment area of 5-6 cm. Information completed in lines 53-55. Although the light is bigger than the area to treat, the photosensitizer is only applied in the nails, so in the rest of normal skin the light does not produce effect.
  • Regarding the incubation time, we use the protocol already adopted by our research group and developed in the following publication, where it is shown that more important than the incubation time, it is the previous nail preparation with 40% urea to enhance the penetration of Metvix.

“Robres P, Aspiroz C, Rezusta A, Gilaberte Y. Usefulness of Photodynamic Therapy in the Management of Onychomycosis. Actas Dermosifiliogr. 2015 Dec;106(10):795-805. English, Spanish. doi: 10.1016/j.ad.2015.08.005. Epub 2015 Oct 1. PMID: 26427737.”

  • Thank you very much for your interesting comment the two other photosensitizers, we have added the published data on curcumin as a photosensitizer in the treatment of onychomycosis.
  • In figure 1, the protocol shown was the one proposed in the conclusions, 3 sessions separated by 1 or 2 weeks of MAL-PDT combined with terbinafine (250 mg/day) for 1 month, followed by another cycle of 3 sessions of MAL-PDT if necessary. This is our preferred protocol if the patient does not deny oral treatment.

Reviewer 2 Report

The manuscript “Combination of photodynamic therapy and oral antifungals for the treatment of onychomycosis.” addresses the question of Photodynamic therapy (PDT) as a therapeutic alternative for onychomycosis. The authors demonstrated improved clinical response (higher cure rates). The work is precedent, although it is preliminary.

There is room for significant improvement:

Major points:

  1. The hypothesis is only partially tested: Line 25“PDT is a therapeutic alternative for onychomycosis, and can be administered either in monotherapy or combined with antifungals, allowing for a reduction in the duration and possible adverse effects of antifungal treatment and achieving higher cure rates…”. It sounds like the authors want to test their hypothesis in two parts: reduction of duration, and higher cure rate. Indeed, The higher cure rates are supported by data presented in Line 173. Yet, the reduction in duration data was not clearly presented, which is again one of the aim of this paper (Line 46 “a reduced duration of systemic antifungal treatment”). Instead, the authors concluded in Line 191 “a reduction in the required dose of systemic antifungals”. Please address the discrepancy in the aim (reduced duration) and the conclusion (reduced dose).
  2. Could the authors further demonstrate the benefit of the combination of PDT+terbinafine? PDT+terbinafine showed a better clinical response (70 %) than PDT alone (53.85) (Line 173). But, with highest response rate of 76% (Line 122) of terbinafine alone, PDT+terbinafine does not seem to outperform terbinafine alone. A justification for such combination therapy is needed.
  3. Figure 2 presents a proposed treatment protocol based on this paper and previous studies. It would be nice for the authors to validate such protocol and support with data.

Author Response

Thank you very much for your smart comments.

All comments have been answered and highlighted in red in the main text.

1.Effectively our hypothesis has two parts, reduction of the duration of antifungal treatments, and higher cure rate. The reduction in the required dose of systemic antifungals is presented in the Results paragraph in lines 90 and 91 Ten patients (50%) received concomitant systemic terbinafine treatment (8 for only 4 weeks and 2 for 12 weeks)” and in Table 1 and Table 2, where we want to highlight that in most patients who need systemic treatment to achieve a complete response, they followed the treatment only for 4 weeks instead of 12 weeks (normal period of treatment time).

  1. We have detected a mistake in line 173, the correct clinical response is 80% instead of 70 %, which is shown in results, in line 93. So, our clinical response is higher than terbinafine alone (80 vs 76%). Furthermore, with the combination therapy we can reduce the treatment with terbinafine to only one month, with the reduction of side effects associated with systemic treatment.
  2. Thank for your comment, we are already working in the validation of this protocol and we are collecting data to support it. Added in lines 184-185.

Reviewer 3 Report

I revised the Manuscript ID: pharmaceuticals-1714021 entitled “Combination of photodynamic therapy and oral antifungals for the treatment of onychomycosis” submitted to Pharmaceutical by Bielsa et al.

The authors reported a study that envolved the use of PDT for treatment of onychomycosis, either in monotherapy or combined with systemic or topical terbinafine, and they achieving higher cure rates than those obtained with either treatment alone.

I enjoyed reviewing this manuscript, which despite being a very synthetic article, seeming not to have been written by academics, mentions all the essential points concerning the intended objectives of this study.

The authors make a good discussion of the results obtained and in the conclusions, they leave open the need for further clinical tests necessary to carry out an optimum protocol for dermatophyte and mold onychomycosis.

I leave only 3 small corrections that contribute to the improvement of this paper.

Minor points:

Line 11 – Change bold “Onychomycosis” to plain.

In line 12 the authors present the meaning of the abbreviation PDT, so it makes no sense that the meaning of PDT appears again at the end of tables 1 and 2 Given the above, I am of opinion that this paper should be accepted after the suggested corrections.

The legend of figure 2 must not be cut by the figure itself.

Given the above, I am of opinion that this paper should be accepted after the suggested corrections.

Author Response

Thank you very much for your comments, I am delighted that you enjoyed the manuscript. All corrections suggested have been adopted and have been corrected and highlighted in red in the main text.

Round 2

Reviewer 2 Report

The authors have suitably edited the manuscript, addressing the reviewer comments and clarifying key elements.